# Development of a SPE-HPLC-PDA Method for the Quantification of Phthalates in Bottled Water and Their Gene Expression Modulation in a Human Intestinal Cell Model

**Vincenzo Ferrone** [1,*] , **Pantaleone Bruni** [1] , **Teresa Catalano** [2] , **Federico Selvaggi** [3] , **Roberto Cotellese** [4,5] , **Giuseppe Carlucci** [1] and **Gitana Maria Aceto** [4,*]

1    Department of Pharmacy, University "G. d'Annunzio" Chieti-Pescara, 66100 Chieti, Italy
2    Department of Clinical and Experimental Medicine, University of Messina, Via Consolare Valeria, 98125 Messina, Italy
3    SS. Annunziata Hospital, ASL2 Lanciano-Vasto-Chieti, Unit of Surgery, 66100 Chieti, Italy
4    Department of Medical, Oral and Biotechnological Sciences, "G. d'Annunzio" University, Chieti-Pescara, Via dei Vestini 31, 66100 Chieti, Italy
5    Villa Serena Foundation for Research, 65013 Città Sant'Angelo, Italy
*    Correspondence: vincenzo.ferrone@unich.it (V.F.); gitana.aceto@unich.it (G.M.A.); Tel.: +39-0871-3554115 (G.M.A.)

**Abstract:** Phthalates are ubiquitous pollutants that are currently classified as endocrine disruptor chemicals causing serious health problems. As contaminants of food and beverages, they come into contact with the epithelium of the intestinal tract. In this work, a SPE-HPLC-PDA method for the determination of phthalates in water from plastic bottles was developed and validated according to the food and drug administration (FDA) guidelines. A chromatographic separation was achieved using a mobile phase consisting of ammonium acetate buffer 10 mM pH 5 (line A) and a mixture of methanol and iso-propanol (50:50 *v/v*, line B) using gradient elution. Several SPE cartridges and different pH values were investigated for this study, evaluating their performance as a function of recovery. Among these parameters, pH 5 combined with the SPE sep pack $C_{18}$ cartridge showed the best performance. Finally, the proposed method was applied to the analysis of real samples, which confirmed the presence of phthalates. A colonic epithelial cell model was used to evaluate the effects of these phthalates at the concentrations found in water from plastic bottles. In cells exposed to phthalates, the increased expression of factors, which control the signaling pathways necessary for intestinal epithelium homeostasis, inflammatory response, and stress was detected. The proposed method falls fully within the limits imposed by the guidelines with precision (RSD%) below 7.1% and accuracy (BIAS%) within −4.2 and +6.1.

**Keywords:** HPLC; pollutions; phthalates; colon disease; Caco-2 cell line; qRT-PCR; gene expression

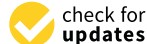



## 1. Introduction

Many industrial chemicals, if not used and disposed of properly, spread to the environment polluting the entire ecosystem. In particular, some plasticizers, also used in packaging, can accumulate in household dust, work environments, and can contaminate food and beverages with the possibility of coming into contact with the epithelium of the intestinal tract [1–3]. The esters of phthalic acid, better known as phthalates (PAEs), are mainly used in the chemical industry as plasticizers to modify the hardness of polymers, such as PVC or used as solvents or fixatives in cosmetics [4]. Among all the used plasticizers 65% are PAEs, and the remaining 35% represent alternative molecules, such as esters of the cyclohexane (DINCH), trimetilates, polymers, aliphatic and epoxy esters, and other molecules of various kinds. The country with the highest consumption of plasticizers is China (42%), followed by Western Europe (14%), the United States (11%), and other countries to a lesser extent [5,6].

Phthalates can be classified into mono- or di-esters of the alcohol that is part of the molecule. They are compounds containing a benzene ring, and one or two ester groups. We can divide them into low molecular weight (LMW) and high molecular weight (HMW) compounds [7]. Since the field of application of PAEs in the chemical industry is vast, the body's exposure to these substances can occur on several fronts, for example from food or drink packaging and by inhalation or direct contact with the skin [8–10]. The main toxic inducing compounds in humans are PAE mono-esters, while glucuronate derivatives are considered mostly non-toxic, there is generally no discrimination between the two categories [11]. Phthalates were counted among the large group of contaminants, which can interfere with metabolism or hormone function and therefore, are called EDCs (Endocrine Disrupting Chemicals) [12–14] In addition, the presence of EDCs in micro(nano)plastics (MPs/NPs) are raising a growing concern for human and ecosystem health. Exposure to these pollutants could also adversely affect gut health and predispose more people to a worsening and/or chronicity of gut diseases [1,15–17]. Therefore, it is important to simplify the ability to detect pollutants, such as PAEs, as well as to evaluate their effects on stable human cell models. Indeed, sample preparation in the analysis of PAEs is very important as their concentration within the matrix is usually below the limits of detection (LOD) and limit of quantification (LOQ) of a standard method. For this reason, one goal of sample preparation is to concentrate the target analytes to be detected and quantified. In recent years, a solid phase extraction and its subsequent miniaturizations have become the main option as a sample preparation technique. SPE is one of the simplest sample preparation techniques for extraction and preconcentration of organic trace contaminants from environmental matrices. It was first introduced in 1970 and has had a great impact in the field of analytical sciences. The efficiency of SPE extraction has been greatly improved with developments over the years in the field of sorbents. When compared to other liquid extraction techniques for organic pollutants, SPE provides a superior performance in analyzing a wide variety of organic pollutants using reduced sample quantities and a lower consumption of polluting solvents [18–20]. Despite the short half-life in tissues, chronic exposure to phthalates can adversely affect the human endocrine system [21]. Their widespread use has therefore made it necessary to develop different methods of analysis. Generally, since phthalates are classified as endogenous destructors, different methods and different sample preparation techniques were developed, ranging from a solid phase microextraction (SPME) to a liquid-liquid dispersive microextraction (DLLME) for their determination. The instrumental techniques with which the PAEs determination is usually carried out are high efficiency liquid chromatography (HPLC) or ultra-high efficiency liquid chromatography (UHPLC) coupled to a UV detector a diode array detector (DAD), and to mass spectrometry. Although liquid chromatography is the preferred technique due to its wide range of applications, gas chromatography coupled with flame ionization detection (FID) or even mass spectrometry also remains a valid alternative in the determination of PAEs [22–27].

In this study, a SPE-HPLC-PDA method was developed which is useful for the analysis of phthalates in water. This method was validated in accordance with the international guidelines. Furthermore, parameters, such as the type of adsorbent material and the sample loading pH were evaluated in order to assess the effects they have on the performance of the SPE. After fine-tuning, the method was successfully applied to the quantization and characterization of six different phthalate molecules in water contained within plastic bottles. Finally, the effects of phthalate pollutants, at the average concentrations found, were evaluated in a stable model of colonic epithelial cells under conditions of phenotypic differentiation.

## 2. Materials and Methods

### 2.1. Chemicals and Selected Phthalates

The Standards used in this study were DiMethyl Phthalate (DMF) (CAS 131-11-3), purity ≥99%, DiEthyl Phthalate (DEF) (CAS 84-66-2), purity 99.5%, DiPropyl Phthalate, (DPF) (CAS 131-16-8), 98% purity, DiButyl Phthalate (DBF) (CAS 84-74-2), 99% purity,

DiIsoButyl Phthalate (DIBF) (CAS 84-69-5), 99% purity, and Di 2 Ethyl Hexyl Phthalate (DEEF) (CAS 117-81-7), 99.7% purity. All phthalates were purchased from Sigma-Aldrich (Milan, Italy). The 98% pure ammonium acetate (CAS 631-61-8) and the monobasic sodium phosphate (CAS 7558-80-7) with $\geq$99% purity, used for the aqueous buffers were purchased from Sigma-Aldrich (Milan, Italy). Acetonitrile (CAS 75-05-8), HPLC purity grade was purchased from Sigma-Aldrich (Milan, Italy), methanol (CAS 67-56-1) HPLC GOLD purity and isopropanol (CAS 67-63-0) with HPLC PLUS grade purity were instead purchased from Carlo Erba Reagenti (Milan, Italy.) The water used was obtained by passing it through an Elix 3 and Milli-Q Academic water purification system (18 m$\Omega$/cm, TOC < 5 ppb) (Millipore, Bedford, MA, USA).

### 2.2. Instrumentation and HPLC Conditions

The chromatographic analysis was performed using a Waters HPLC system, consisting of a Rheodyne model 7725i injector with a 20 µL loop and a Waters 600 pump. The instrumentation is completed by a 2996 PDA diode array detector. The software used for the data acquisition was Empower v.2 (Waters, Milford, MA, USA). An analytical balance "Precisa" model XT120A was used while a Labsonic ultrasonic bath (FALC, Milan, Italy) was used for sonication and an Eppendorf Centrifugate 5804 centrifuge was used to centrifuge the samples. The SPE cartridges used in this study were Bond Elute Plexa from Agilent, Strata–X and Strata $C_{18}$-E from Phenomenex, Sep-Pak Vac from Waters, Evolute Express ABN from Biotage, Bakerbond Octadecyl from J.T. Bakers and Macherey-Nagel's Chromabond $C_{18}$. The multiple extraction system used in the SPE is a Visiprep 12 solid Phase Extraction Vacuum Manifold. Filtration was performed with a 5 mL Micro-Mate filter syringe, using 0.45 µm pore size syringe filters.

HPLC separation of the investigated phthalates was obtained using a Poroshell 120 $C_{18}$ (150 × 4.6 mm I.D. 4 µm particle size) protected by a Fast guard (5 × 4.6 mm I.D.) at the temperature of 20 $\pm$ 1 °C using a column heater. The mobile phase was ammonium acetate buffer 10 mM pH 5 (line A) and a mixture of methanol and isopropanol (50:50 *v/v*, line B) using gradient elution. A linear gradient was used for the separation of the PAEs; the initial composition of the mobile phase was 65% A, then in 20 min the percentage of A decreased to 8% and remained constant for 10 min, it then increased back to 65% after 1 min followed by 7 min of re-equilibration. For the quantitative analysis of the DMF, DEF, DPF, DBF, DIBF, and DEEF, the wavelengths used were 226, 223, 221, 223, 220, and 272 nm, respectively. The solvents were filtered before use through a 0.45 µm WTP membrane, while ammonium acetate solution was filtered through a WCN 0.5 µm membrane and 20 µL was injected into the system. The total run time was 38 min.

### 2.3. Preparation of Standard Solutions

The stock solutions of the studied PAEs (DMF, DEF, DPF, DBF, DIBF, and DEEF), were initially prepared at a concentration of 4 mg/mL by withdrawing the exact amount of the standard with a micropipette and bringing to the required volume in 25 mL volumetric flasks with acetonitrile. The stock solutions were diluted to obtain the working solution using water and acetonitrile 75/25 *v/v*. The solutions used for the calibration curve were obtained by taking an appropriate amount of the working solution to obtain samples with nominal concentrations of 0.01, 0.05, 0.1, 0.25, 0.75, 2.5, 5.0, and 10.0 µg/mL. Quality control samples (QCs) were prepared independently from calibrators in the same way as previously reported to obtain the final concentration of 0.02 (QCL), 0.50 (QCM), and 7.5 (QCH) µg/mL.

### 2.4. Sample Preparation

Samples were transferred to glass bottles and stored in a refrigerator at 4 °C until analysis. Samples were loaded onto a conditioned Sep-pack $C_{18}$ cartridges and drawn through on an Visiprep 12 solid Phase Extraction Vacuum Manifold using a vacuum. They were conditioned with 2 × 1 mL of acetonitrile followed by 2 × 1 mL of sodium acetate

buffer 10 mM (pH 5). After loading the samples, they were consecutively washed with $2 \times 1$ mL of water. Analytes were eluted with 1mL of methanol. The eluent was further evaporated to dryness under a nitrogen stream using a drying attachment apparatus (Supelco, Bellefonte, PA, USA) at room temperature and re-dissolved in a 200 µL mixture at the initial composition of the mobile phase, vortexed for 1 min, filtered on a Phenex-PTFE (4 mm, 0.45 µm) filters, and 20 µL were injected into the HPLC system.

## 2.5. Human Colon Cell Culture and Treatments

For this study we used a Caco-2 colon cancer cell line with different degrees of differentiation: Caco-2 as a model for colon epithelium phenotype at differentiation (15 days after seeding) and a low degree of differentiation (2 days after seeding) [28]. The lines were obtained from American Type Culture Collection (ATCC) (Manassas, VA, USA). Caco-2 cells were cultured at 37 °C in DMEM medium containing 10% fetal bovine serum (FBS), 100 U/mL penicillin/streptomycin, and 2 mM L-glutamine (EuroClone, Pero, MI, Italy). For these experiments, cells were treated with phthalates for 24-h exposure at different concentrations 10 ng/mL, 100 ng/mL, and 500 ng/mL. PAEs exposure was performed in two modes, namely, in the undifferentiated phase and in the differentiating phase.

## 2.6. Cell Viability and Metabolic Assay

The interference of PAEs with cell viability and metabolic activity was assessed by a colorimetric assay. Caco-2 cells were grown in 96-well plates at a concentration of $1.0 \times 10^4$ cells/well. Cells were exposed to increasing concentrations of low dose of PAEs (10 ng/mL, 100 ng/mL, and 500 ng/mL) for 24 h. This was followed by incubation with 10 µL/well of 2-[2-methoxy-4-nitrophenyl]-3-[4-nitrophenyl]-5-[2,4-disulphophenyl]-2H-tetrazolium, monosodium salt (MTS) assay (Promega, Madison, WI, USA) at 37 °C for 1 h. the absorbance was measured at 490 nm with a Synergy H1 microplate reader (BioTek Instruments Inc., Winooski, VT, USA). For each experimental condition, five repetitions were performed, and the results were validated in two independent experiments. The significance of the obtained data was considered with $p \leq 0.05$.

## 2.7. RNA Extraction, Reverse Transcription, and Real-Time Quantitative Polymerase Chain Reaction (qRT-PCR)

Total RNA was isolated from independent cultures of differentiated (16 days) and nondifferentiated (3 days) Caco-2 cells exposed to each phthalate at a concentration of 100 ng/mL for 24 h. EuroGold TriFast reagent (EuroClone) was used for the extraction according to the manufacturer's instructions. The RNA samples were assessed for purity and quantified using a Nanodrop 1000 Spectrophotometer (Thermo Fisher Scientific, Waltham, MA, USA). The synthesis of the complementary DNA (cDNA) was performed employing the GoTaq® 2 Step RT-qPCR Kit (Promega) according to the manufacturer's instructions. The mRNA levels were evaluated using SYBR Green quantitative real-time PCR (qRT-PCR) analysis using StepOne™ 2.0 (Applied Biosystems, Thermo Fisher Scientific). The Data were analyzed using the comparative Ct method and were graphically indicated as $2^{-\Delta\Delta Ct}$ + SD. In accordance with the method, the mRNA amounts of the target genes were normalized by the ratio on the median value of the endogenous housekeeping gene glucuronidase-beta (*GUSB*) obtained in treated cells vs. untreated cells. Target and reference genes were amplified in triplicate as described in Catalano et al., 2021 [29]. The mRNA expression levels were analyzed for the E-cadherin calcium-dependent adhesion protein (*CDH1*), essential to the formation of the intestinal barrier [30]; the Lymphoid Enhancer-Binding Factor 1 (*LEF1*) a transcription factor, which is involved in the canonical Wnt/β-catenin signaling pathway [31]; the *AP-1* transcription factor subunit of Jun Proto-Oncogene, involved in the non-canonical Wnt/β-catenin signaling pathway [29];and the *P65/RELA* Proto-Oncogene NF-kB transcription factor subunit [32]. Sequences of the oligonucleotides used as qRT-PCR primers will be provided upon request. The results were subjected to *t*-test analysis and the statistical significance was set at $p \leq 0.05$.

## 3. Results

### 3.1. Development of SPE Procedure

The final aim of a proper sample preparation is to transform a matrix containing analytes into an easy-to-analyze sample. The greatest difficulty of the simultaneous analysis of phthalates is the choice of the best SPE sorbent to give an acceptable recovery for all analytes. In the preliminary experiments, seven different SPE materials for the extraction of the investigated phthalates were tested: Bond Elute Plexa from Agilent (30 mg/1 mL), Strata–X (30 mg/1 mL) and Strata $C_{18}$-E (100 mg/1 mL) from Phenomenex, Sep-Pak Vac (100 mg/1 mL) from Waters, Evolute Express ABN (25 mg/1 mL) from Biotage, Baker-bond Octadecyl (100 mg/1 mL) from J.T. Bakers, and Macherey-Nagel's Chromabond $C_{18}$ (100 mg/1 mL). The sorbent was chosen on the basis of the physicochemical properties of the phthalates. The dominant retention mechanism in all cartridges is reverse phase (RP). The reverse phase was the obvious choice as a retention mechanism for the solid phase extraction of phthalates. Cartridges were compared using recovery as a response. The experiments were performed in triplicate on standard solutions having a concentration of 1.0 μg/mL for each phthalate. Results are presented in Figure 1. Furthermore, different pH (from 2 to 7) of the samples were investigated. The best results were obtained when pH 5.0 was used, represented in Figure 2.

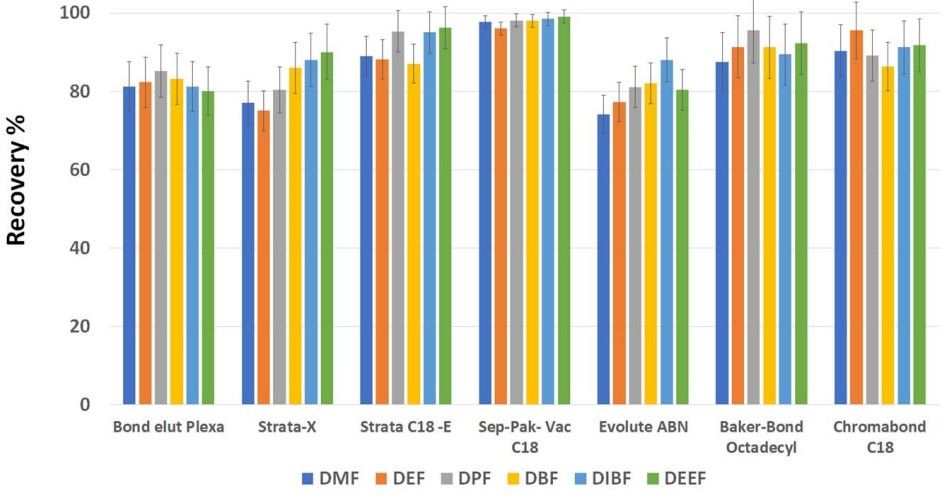

**Figure 1.** Comparison of SPE cartridges on the recovery of standard solutions containing the investigated phthalates at a concentration of 1.0 μg/mL.

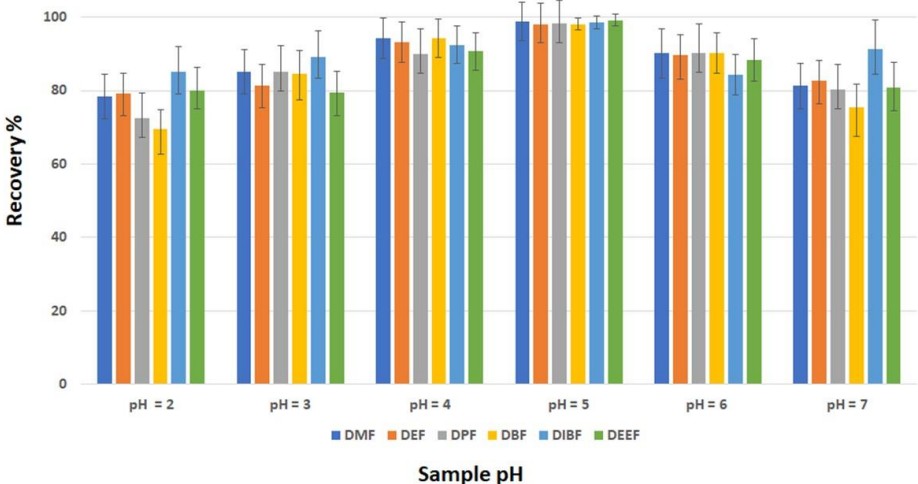

**Figure 2.** Effect of sample pH on the extraction of phthalates for the SPE-HPLC-PDA method.

### 3.2. Method Development

The proposed method was validated according to the FDA guidelines (F.D.A. Guidance for Industry: Bioanalytical Method Validation, U.S. Department of Health and Human Services, Food and Drug Administration, Center for Drug Evaluation and Research & Center for Veterinary Medicine, 2003.) The selectivity of the method was tested by extracting six different batches of blank matrix. The results suggested the absence of chromatographic interferences between the interferents and the analytes. Furthermore, the analytes were well resolved. Linearity was evaluated by constructing a calibration curve for each analyte in the range of 0.01–10 µg/mL. A calibration curve was constructed via a least square linear regression using concentration as an independent variable and the response (area) as the dependent variable. The statistical analysis of the concentration–response ratios proved in all cases that the linear correlation was the best model in the concentration range studied, with a mean correlation coefficient ($r^2$) > 0.9990. Linearity was also confirmed by the analysis of the back calculated concentration of calibrators. Limits of detection (LOD) and quantification (LOQ) were evaluated as signal-to-noise ratios. The limit of detection and quantification was 0.003 µg/mL and 0.01 µg/mL for all the PAEs, respectively. Precision and accuracy were assessed by the analysis of three batches of quality control samples (QCs) at three concentrations in triplicate and for five consecutive days (n = 5) as shown in Table 1. Results suggested that the proposed method satisfies the FDA guidelines because the precision (RSD%) is below 7.1% and the accuracy (BIAS%) is within −4.2 and +6.1. Sep Pack $C_{18}$, among the other sorbents, showed the best recoveries. Recovery was evaluated using spiking blank samples and comparing the ratio spiked blank extract to the standard solution at the same concentration.

**Table 1.** Precision and accuracy for the proposed SPE-HPLC-PDA method.

| ANALYTE | Precision (RSD%) | | Accuracy (BIAS%) | |
|---|---|---|---|---|
| | Inter-Day | Intra-Day | Inter-Day | Intra-Day |
| DMF | 1.7 | 1.8 | −2.1 | +6.1 |
| | 2.5 | 1.6 | +3.4 | +3.2 |
| | 1.9 | 3.2 | −4.2 | −2.7 |
| DEF | 2.4 | 2.9 | +5.4 | −0.9 |
| | 3.1 | 3.7 | −2.0 | −0.8 |
| | 5.2 | 4.8 | −1.7 | +4.8 |
| DPF | 1.5 | 3.4 | −1.5 | +4.9 |
| | 6.3 | 3.4 | +3.9 | −0.1 |
| | 4.1 | 6.1 | +4.5 | +5.5 |
| DBF | 5.3 | 5.5 | +3.9 | +3.2 |
| | 1.5 | 5.8 | +4.4 | +0.9 |
| | 1.8 | 6.7 | −1.8 | +0.5 |
| DIBF | 5.5 | 7.0 | −1.9 | −1.9 |
| | 2.9 | 6.5 | +5.5 | +5.9 |
| | 3.4 | 5.9 | +6.0 | +0.7 |
| DEEF | 7.1 | 6.3 | −2.0 | −1.0 |
| | 6.4 | 6.7 | +1.9 | +4.9 |
| | 4.9 | 5.7 | −0.5 | −0.8 |

### 3.3. Analysis of Bottled Water Samples

Samples were collected from several local markets. The water from 10 different brands, in plastic bottles, was investigated. The results showed that in all the water samples at least one of the analytes investigated was quantified, while in some samples two phthalates were found, and in one sample three were found. The results of the analyzes are shown in Table 2. Among the PAEs investigated, DPF and DEEF were those found at the highest concentrations, while DBF or DIBF were found or detected in all the plastic bottles.

**Table 2.** Concentration of investigated phthalates in bottled water (µg/mL).

| Samples | DMF | DEF | DPF | DBF | DIBF | DEEF |
|---------|-----|-----|-----|-----|------|------|
| #1 | D | ND | ND | ND | 0.026 | 0.014 |
| #2 | D | ND | 0.043 | ND | 0.021 | ND |
| #3 | ND | ND | ND | ND | 0.034 | 0.010 |
| #4 | ND | D | 0.019 | D | 0.022 | ND |
| #5 | 0.012 | ND | 0.045 | D | D | ND |
| #6 | ND | ND | 0.032 | ND | 0.011 | D |
| #7 | ND | ND | 0.074 | 0.016 | ND | 0.028 |
| #8 | ND | ND | ND | ND | 0.033 | D |
| #9 | D | 0.052 | ND | D | D | 0.011 |
| #10 | ND | D | 0.041 | ND | D | 0.023 |

ND: not detected (concentration < LOD). D: detected (concentration < LOQ).

### 3.4. Cell Viability and Metabolic Assay

We analyzed the bioactivity of the three most frequently detected phthalates in the water samples tested (DIBF, DEEF, and DPF) see Table 2. Their effect on cell viability and metabolism was tested by the MTS assay in the two-mode Caco-2 human colon model with undifferentiated and differentiated phenotypes. The exposure to increasing concentrations of phthalates (10 ng/mL, 100 ng/mL, and 500 ng/mL) induced slight positive changes on the viability of proliferating Caco-2 cells (Figure 3), whereas in the differentiated cells a tendentially negative effect was observed by BEEF and DPF (Figure 4). Specifically, DPF at concentrations of 100 ng/mL and 500 ng/mL induced a significant reduction in viability (**: $p = 0.005$ and *: $p = 0.046$, respectively).

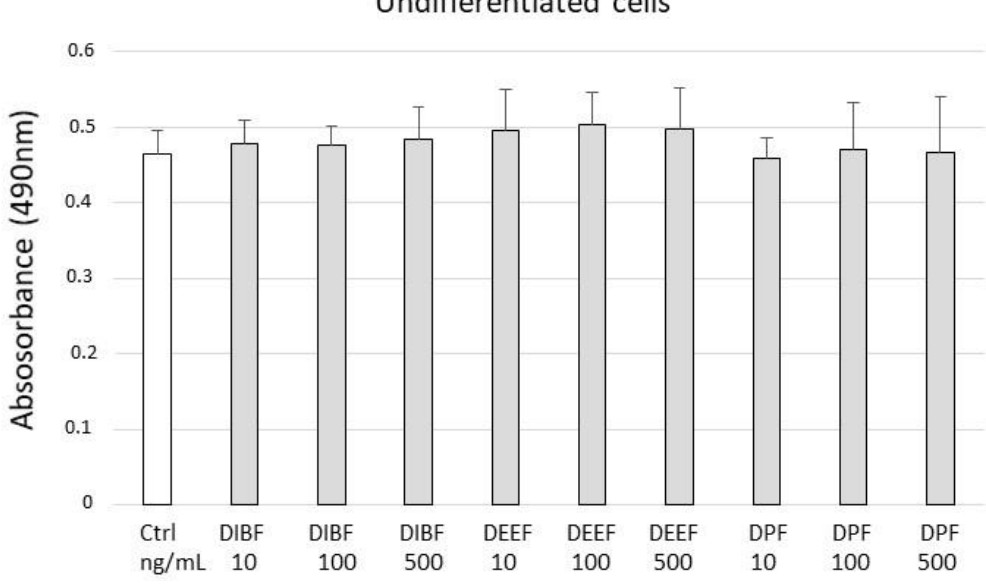

**Figure 3.** Caco-2 cells treated for 24 h with DIBF, DEEF, and DPF in exponential growth phase, proliferating (at day 3 after seeding). No significant changes in cell viability were observed. The evaluation was performed by MTS assay, as an indirect measure of mitochondrial metabolic capacity.

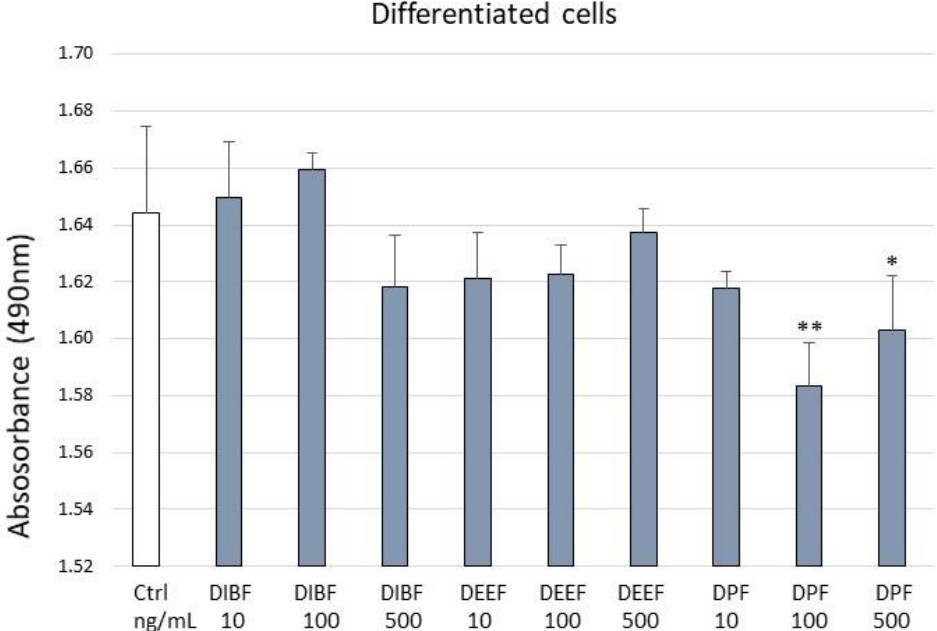

**Figure 4.** Differentiated Caco-2 cells treated for 24 h with DIBF, DEEF, and DPF (10 ng/mL, 100 ng/mL, 500 ng/mL). In the differentiated growth phase Caco-2 cells exposed for 24 h from day 15 of seeding, a slight decline in viability was observed following treatment with phthalates at 500 ng/mL. While exposure to DPF at both 100 ng/mL and 500 ng/mL resulted in a significant reduction in the cell viability with $p = 0.005$ and $p = 0.046$ respectively. * $p < 0.05$, ** $p < 0.01$.

*3.5. Gene Expression Assay by qRT-PCR*

To assess whether exposure to PAEs might affect the modulation of gene expression in colonic epithelium, we analyzed differentiated and undifferentiated Caco-2 cells, the mRNA expression of E-cadherin, and three transcription factors that regulate pathological aspects of the colon [29–32]. The model line under these two conditions was treated for 24 h with DIBF, DEEF, and DPF at 100 ng/mL. This model was investigated on DIBF, DEEF, and DPF since the analysis of the real samples underlined their presence in most of the samples analyzed. The effects were compared with those from the untreated cells (Figure 5). Distinct biological effects were found in colon epithelial cells following the treatments. In particular, a significant reduction in gene expression of E-cadherin (CDH1 gene), transcription factors P65/RELA and JUN/AP1 was observed in non-differentiated cells after treatment with BEEF 100 ng/mL. While DPF induced the reduction in P65/RELA and an increase in JUN/AP1. In differentiated cells, DPF induced a significant increase in expression of both E-cadherin and the transcription factors, whereas BEEF alone reduced the expression of JUN/AP1 (Figure 5).

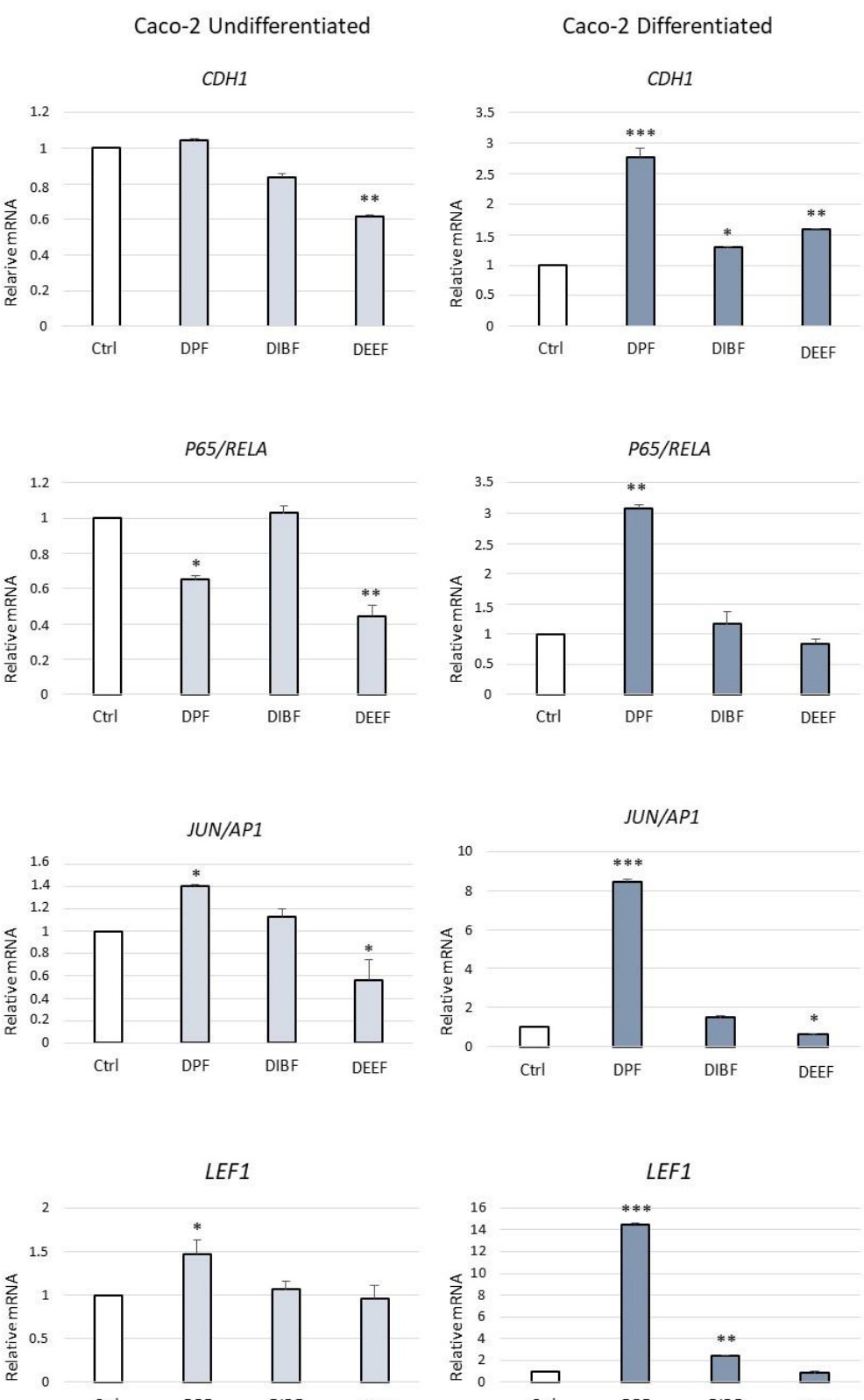

**Figure 5.** Gene expression modulation under PAE treatments in Caco-2 cells. Gene expression was analyzed by Real Time-qPCR. The histograms represented normalized data with GUSB gene. CDH1 (E-cadherin); P65/RELA (P65/RELA Proto-Oncogene NF-kB transcription factor subunit); JUN/AP1 (AP-1 transcription factor subunit of Jun Proto-Oncogene); LEF1 (Lymphoid Enhancer-Binding Factor 1). The histogram represented normalized data with GUSB gene. The results showed the average of three independent experiments. * $p < 0.05$, ** $p < 0.01$ and *** $p < 0.001$ treated vs. untreated cells.

## 4. Discussion

### 4.1. Optimization of Chromatographic Conditions

The identification of the best experimental condition for the chromatographic separation of different analytes is a task that requires a considerable number of experiments. In this work, different mobile phases at different pH values (2.5, 5.0, and 7.5) were tested. Using pH 2.5 and 7.5 (10 mM phosphate buffer) it was noted that the noise increased, decreasing the sensitivity of the method, while using pH 5.0 a lower noise was obtained. Subsequently, the type of elution: isocratic or gradient, was evaluated. It is universally recognized that isocratic elution is preferable to gradient elution if possible; however, given the nature of the analytes and their different lipophilicity it was not possible to use isocratic elution. Different gradients were explored with the aim of separating as much as possible the butyl-phthalate from the isobutyl phthalate and the gradient reported in Section 2.2 was the only gradient able to separate them. In addition to the various gradients, several columns were also investigated, a Gemini column (150 × 4.6 5μm particle size) was tried, which proved to be unsuitable as it produced very large peaks, and a Cortecs column (150 × 4.6 5μm particle size), which did not produce the separation of the two PAEs with similar retention times (DBF and DIBF). Poroshell 120 C$_{18}$ (150 × 4.6 mm I.D. 4 μm) had the best performance and was used for the separation of the phthalates. Chromatograms of the blank extract and the blank spiked extract is shown in Figure 6, while a sample extract subjected to the proposed SPE-HPLC-PDA method is shown in Figure 7.

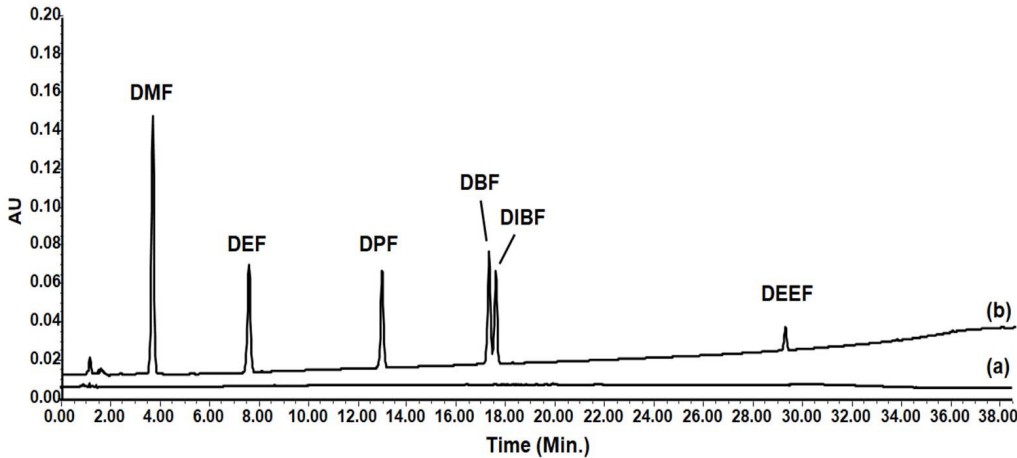

**Figure 6.** HPLC-PDA chromatograms of blank extract (a) and blank spiked extract (b) with the investigated phthalates at the concentration of 0.75 μg/mL.

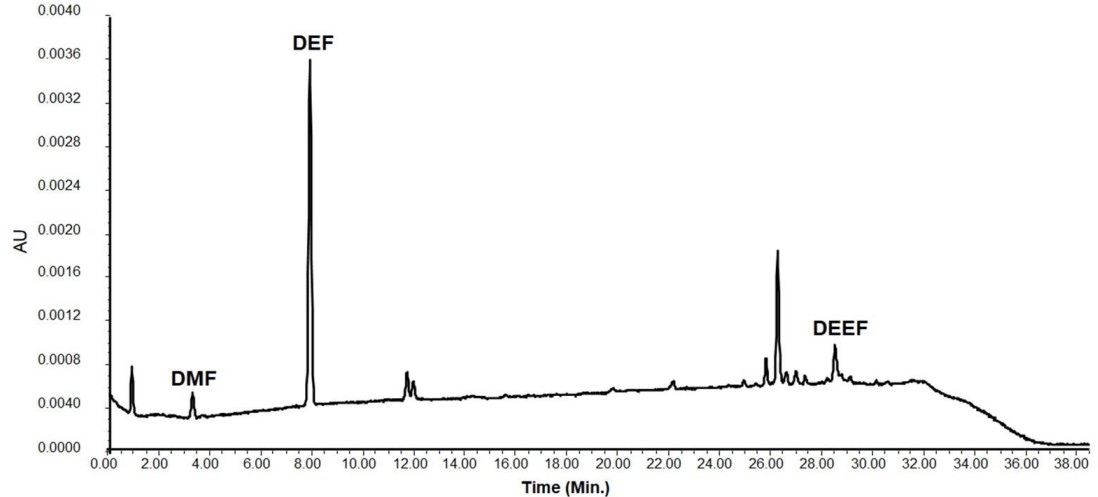

**Figure 7.** Bottled water sample extract subjected to the proposed SPE-HPLC-PDA method.

### 4.2. HPLC-PDA Comparison with Other Methods

The proposed method was compared with other methods present in the literature [33–37]. The comparison is reported in Table 3. As reported in Table 3, the proposed HPLC-PDA method has several advantages, including low LOQs and higher recoveries. The results revealed a good applicability of the proposed in the analysis of PAE in water samples. In the literature there are methods which provide the liquid-liquid extraction as a sample preparation technique. This technique when compared with the solid phase extraction involves a higher consumption of environmentally toxic organic solvents. Furthermore, the solid-phase extraction when compared to the liquid-liquid extraction provides more precise and accurate results as it is subject to less variability. Liquid-Dispersive Extraction with Solidification of Floating Droplets (DLLME-SFO) is a fairly recent sample preparation technique, it can be considered an evolution of the liquid-dispersive microextraction introduced by Rezaee et al. [38]. This technique reduces the use of organic solvents and offers high preconcentration factors; however, it has disadvantages, such as the formation of microemulsions and difficulty in recovering the floating droplets, which often coincided with a loss in precision and accuracy.

**Table 3.** Comparison with methods reported in the literature.

| Analytes | Sample Preparation | Instrumentation | Limit of Quantification (μg/mL) | Ref |
|---|---|---|---|---|
| DEF | SLE [1] | HPLC-PDA | 0.07 | [33] |
| DEEF | LLE [2] | HPLC-PDA | 0.05 | [34] |
| DMF, DEF, DBF, DEEF | LLE | HPLC-PDA | 0.64 | [35] |
| DEF, DIBF, DEEF | DLLME-SFO [3] | HPLC-PDA | 0.10 | [36] |
| DMF, DEF, DEEF | ASE [4] | HPLC-PDA | 1.00 | [37] |
| DMF, DEF, DPF, DBF, DIBF, DEEF | SPE [5] | HPLC-PDA | 0.01 | proposed method |

[1] Solid Liquid extraction; [2] Liquid Liquid extraction; [3] Dispersive Liquid Liquid Microextraction with Solidification of the Floating Organic droplets; [4] Accelerated Solvent Extraction; [5] Solid Phase extraction.

### 4.3. Effects of Phthalates on Bowel Epithelial Cells

Our observations present the first evidence that low concentrations of PAEs, found in aqueous matrices, are able to modulate the biological Caco-2 system by altering gene expression in both proliferating non-differentiated and differentiated cells. The Caco-2 line was derived from colorectal adenocarcinoma developed in the 1970s. Its use as an in vitro model has been widely applied in pharmacology, nutrition, and microbiology. Indeed, it can function as undifferentiated cells of the large intestine or if kept continuously in culture beyond 12 days, it can spontaneously differentiate to resemble a small intestine-like phenotype with enterocyte-like absorptive properties [28,39]. In particular, we evaluated the molecules involved in colonic tissue physiopathology. Indeed, E-cadherin plays a key role in colonic physiology and pathology. It is a major component of epithelial adhesion junctions, which are essential for tissue development, differentiation, and maintenance. It is also crucial for the formation of tissue barriers, a basic function of epithelial tissues. The colon or large intestine is lined by an epithelial monolayer that comprises an E-cadherin-dependent barrier, which is critical for organ homeostasis. Impairment of the colonic epithelium barrier leads to inflammation and fibrosis. Loss of E-cadherin expression is commonly observed in gastrointestinal cancers [40]. It is also considered a tumor suppressor in the colon, mainly because of its function in opposing Wnt signaling, the predominant driver of colon tumorigenesis [31]. In addition to these roles, some recent studies have described E-cadherin as a signaling hub that may regulate cell behavior in response to intra- and extra-cellular stimuli. Interestingly, these recent findings also reveal that, in

some cancer types, its overexpression may promote tumor progression [30]. In Caco-2 differentiated cells, gene expression data shows an upregulation of transcription factors *LEF1, JUN/AP1,* and *P65/RELA* (Figure 5), whose functions can be altered by the chronic exposure to PAEs, resulting in a predisposition to bowel disease and cancer. Indeed, at the early stage of colon carcinogenesis, the evolution from adenoma to adenocarcinoma in human tissue samples, LEF1 expression has been reported increased and has been associated with nuclear accumulation of β-catenin [30]. Moreover, in the initiation and progression of colitis, RELA expression is highly correlated with NF-κB inflammatory bowel diseases [41,42]. Although Caco-2 cells were found to express a large number of enzymes and transporter proteins found in normal human intestinal epithelium, variations obviously exist between the gene expression profiles of transformed epithelial cell lines, such as Caco-2 and normal human intestinal epithelium [43]. However, intestinal epithelial cell models, such as the Caco-2 model, have many advantages due to their simplicity and reproducibility that allows for a comparison of results between laboratories. In addition, investigating the biological effect of a pollutant in an in vitro model enables the study of molecular mechanisms that might be more difficult to address in vivo.

## 5. Conclusions

In this work, a method involving solid-phase extraction was developed and applied for the quantitative analysis of phthalates in water in plastic bottles. The analyses confirmed the presence of one or more of these contaminants. Initially, a comparison was made between the most widely used types of SPE cartridges, which showed that Sep pack C18 had the best extraction performance for phthalates. The method was then validated in accordance with the FDA guidelines for validation of bioanalytical methods. All the parameters that were studied fell well within those imposed. The developed method was finally applied to the analysis of phthalates in bottled water samples. The SPE-HPLC-PDA method, proposed among the methods currently present in the literature, has shown a greater sensitivity in the quantification of phthalates. In addition, it is accessible, versatile, inexpensive with a good performance for its application. This method of analysis can be further applied to the identification of contaminants that come into contact with the gastrointestinal mucosa. Therefore, in this study we proposed a biological model of intestinal mucosa in which to evaluate the activity of pollutants. Phthalate concentrations detected in water samples were able to alter the expression of gene transcription factors in the intestinal cell model. Modulation of the analyzed markers could drive the inflammatory response and renewal processes of the intestinal epithelium. However, further studies will be needed to further investigate the pathophysiological effects of phthalates and their possible implications in intestinal diseases.

**Author Contributions:** Conceptualization, V.F. and G.C.; methodology, V.F. and G.M.A.; validation, V.F., P.B. and G.M.A.; formal analysis, V.F. and G.M.A.; investigation, V.F.; resources, R.C.; data curation, V.F.; writing—original draft preparation, V.F. writing—review and editing, T.C. and F.S.; visualization, G.M.A.; supervision, G.M.A.; project administration, G.M.A.; funding acquisition, R.C. and G.C. All authors have read and agreed to the published version of the manuscript.

**Funding:** This research was funded by the "G. d'Annunzio" University of Chieti-Pescara (Fondi di Ateneo per la Ricerca—F.A.R.) of G.C. and R.C.

**Institutional Review Board Statement:** Not applicable because the study does not involve human subjects.

**Informed Consent Statement:** Not applicable because the study does not involve human subjects.

**Data Availability Statement:** The datasets used and/or analyzed during the current study are available from the corresponding authors upon reasonable request.

**Conflicts of Interest:** The authors declare no conflict of interest.

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
