# Peer review of "Development of a SPE-HPLC-PDA Method for the Quantification of Phthalates in Bottled Water and Their Gene Expression Modulation in a Human Intestinal Cell Model"

_processes, doi:10.3390/pr11010045_

Round 1
Reviewer 1 Report
The present manuscript entitled "Development of a SPE-HPLC-PDA method for the quantification of phthalates in bottled water and assessment of their ability to modulate gene expression in human colonic epithelium model" by Vincenzo Ferrone, Pantaleone Bruni, Teresa Catalano, Federico Selvaggi, Roberto Cotellese, Giuseppe Carlucci, and Gitana M. Maria Aceto (processes-2111945) is written correctly and has a good structure; moreover, it has all the necessary parts. The article is interesting from an analytical and biomedical point of view; therefore, it should interest the reader. I proposed improvements in the method description and with a presentation of figures. The paper meets Processes' requirements, and I recommend the article for publication in Processes following the common editing stage. My current decision is a minor revision. More specific comments and observations are presented below.
1. "Real samples" are mentioned. All samples are real. A better term would be "natural" or "biological".
2. RSD expressed as a percentage is the coefficient of variation (CV).
3. Please correct the typos in the text.
4. Page 2, lines 80-84. This paragraph should be expanded, and references added.
5. "C18" is written once in the index and once normally.
6. Units. Please check the record of units and unify them (“l” or “L”). Units are also written sometimes with the value and sometimes separately.
7. Page 3, line 127. The maximum wavelengths can be listed.
8. Page 3, line 136. Specific concentrations should be listed instead of a range.
9. The axes in the graphs should be more visible.
11. Section 3.2. What can be done in the event of strong interference effects? How would you deal with them? What types of interference effects could occur?
12. The notation "CaCo2" should be standardized in the text. There are currently different entries.
13. Figure 5 is missing in the manuscript, or there is an error in the numbering of figures.
14. Figure 7. Axes should contain name and unit (The same for Fig. 8). Mark (a) and (b) in the drawing caption.
15. Table 3. Please explain what the LOQ value refers to. Is it also possible to add LOQ values and a linear range to the table?
16. Does the developed method have disadvantages?
17. Appropriate tools should be used to best characterize the method when developing a new approach (e.g., AGREE- Analytical GREEnness Metric Approach or RGB model).
18. Conclusions. Please clearly highlight the most important advantage.
I hope that the comments presented will help improve the article.
Author Response
1° Review Report (Round 1)
comments:
The present manuscript entitled "Development of a SPE-HPLC-PDA method for the quantification of phthalates in bottled water and assessment of their ability to modulate gene expression in human colonic epithelium model" by Vincenzo Ferrone, Pantaleone Bruni, Teresa Catalano, Federico Selvaggi, Roberto Cotellese, Giuseppe Carlucci, and Gitana M. Maria Aceto (processes-2111945) is written correctly and has a good structure; moreover, it has all the necessary parts. The article is interesting from an analytical and biomedical point of view; therefore, it should interest the reader. I proposed improvements in the method description and with a presentation of figures. The paper meets Processes' requirements, and I recommend the article for publication in Processes following the common editing stage. My current decision is a minor revision. More specific comments and observations are presented below.
Authors response:
We thank the reviewer for appreciating our work. His careful reading of the manuscript and his comments will help us to improve it. Please find below a detailed point-by-point response to all comments.
Point 1. "Real samples" are mentioned. All samples are real. A better term would be "natural" or "biological".
Response 1. Indeed, this term "Real samples" can confuse the readers, we have used the term "real" in order not to confuse these samples from those obtained by adding a phthalate-free water used in the development phases of the method. To avoid misunderstandings, we have used the term "bottled water samples" instead of real.
Point 2. RSD expressed as a percentage is the coefficient of variation (CV).
Response 2. The authors agree with the referee that RSD% is also called CV. However, following the FDA guidelines, which are mentioned in the text, where the results must be reported in RSD%, we prefer to leave the term relative standard deviation percentage (RSD%) instead of coefficient of variation (CV) to avoid misunderstandings
Point 3. Please correct the typos in the text.
Response 3. The manuscript has been corrected and the typos have been removed.
Point 4. Page 2, lines 80-84. This paragraph should be expanded, and references added.
Response 4. As suggested, we have expanded the paragraph and added new references.
Point 5. "C18" is written once in the index and once normally.
Response 5. The term "C18" has been standardized throughout the text.
Point 6. Units. Please check the record of units and unify them (“l” or “L”). Units are also written sometimes with the value and sometimes separately.
Response 6. The authors have checked and harmonized throughout the text.
Point 7. Page 3, line 127. The maximum wavelengths can be listed.
Response 7. The wavelengths used in the quantitative analysis have been made explicit in the text.
Point 8. Page 3, line 136. Specific concentrations should be listed instead of a range.
Response 8. Thanks for your suggestion, we appreciate it. For a better understanding of the work the concentrations of the working solutions have been made explicit.
Point 9. The axes in the graphs should be more visible.
Response 9. We have re-edited the figures in order to make the axes more readable.
Point 11. Section 3.2. What can be done in the event of strong interference effects? How would you deal with them? What types of interference effects could occur?
Response 11. As required by the guidelines, a method cannot be validated if there is a signal at the retention time of an analyte equal to 20% of the signal obtained at the lowest concentration (LOQ). Indeed, if present, the composition of the mobile phase or gradient should be modified in order to separate the interferents from the analytes at their base. Another alternative could be to explore a different retention mechanism using a column with a different stationary phase. Ultimately, if this is not possible, the sample preparation should be modified and made even more selective. Fortunately, in the proposed method there were no interferences coming from the matrix at the retention time of the phthalates.
Point 12. The notation "CaCo2" should be standardized in the text. There are currently different entries.
Response 12. We rechecked and corrected the notation of Caco-2 cells name.
Point 13. Figure 5 is missing in the manuscript, or there is an error in the numbering of figures.
Response 13. The numbering of the figures has been corrected.
Point 14. Figure 7. Axes should contain name and unit (The same for Fig. 8). Mark (a) and (b) in the drawing caption.
Response 14. Thanks for the suggestion, it was a typo. The figure has been re-edited according to the suggestions of the referee.
Point 15. Table 3. Please explain what the LOQ value refers to. Is it also possible to add LOQ values and a linear range to the table?
Response 15. The LOQ refers to the limit of quantification of the proposed methods. In some of the cited works the range is missing and would not provide additional information. The authors think that the focus should be on improving the sensitivity of the methods as highly concentrated samples can always be diluted before being analysed.
Point 16. Does the developed method have disadvantages?
Response 16. Obviously, it is a method that cannot be compared with mass spectrometry-based techniques that can obtain equal or superior performance in sensitivity without the need for the development of sample preparation techniques. However, these techniques are very expensive, require highly specialized technicians and are not available to everyone. This method, on the other hand, is versatile, economical and has good performance for its application.
Point 17. Appropriate tools should be used to best characterize the method when developing a new approach (e.g., AGREE- Analytical GREEnness Metric Approach or RGB model).
Response 17. The authors agree with the referee's statement on the need to add tools to better characterize an analytical method. These new tools for the evaluation of analytical methods are increasingly coming to the attention of the scientific community, as it should be given the lack of resources. However, throughout the text we have not claimed the greenness of our method as there are steps that cannot be avoided such as chromatography and solid phase extraction. We are hopeful that technological developments will allow the use of solvents that are certainly safer from an environmental point of view. We agree with the referee that these tools are important because they can revolutionize the approach to the development of analytical methods, but like any revolution, it needs time and technological developments to be applied. For this reason, we have decided not to apply the models suggested by the referee AGREE- Analytical GREEnness Metric Approach or RGB model."
Point 18. Conclusions. Please clearly highlight the most important advantage.
Response 18. As suggested, we have expanded the conclusion paragraph, highlighting the most important advantages of the SPE-HPLC-PDA method for the quantification of phthalates in bottled water.

Reviewer 2 Report
The work presented in this manuscript is very interesting, there are some limitations which should be removed before acceptance for publication. The comments are given in word file attached below

Author Response
2° Review Report (Round 1)
comments:
The authors have reported the development of SPE-HPLC-PDA method for the quantification of phthalates in bottled water. There are several limitations in this manuscripts which should be removed before accepting for publications.
Authors response:
We thank the reviewer for his careful reading of the manuscript and for his constructive comments to improve it.
The authors will endeavor to improve the article by following the reviewer's requests.
Below is a detailed point-by-point response to all comments.
Point 1. Title: Development of SPE-HPLC-PDA method for the quantification of phthalates in bottled water and assessment of their ability to modulate gene expression in human colonic epithelium model.
The title is very long and confusing. ---- their ability to modulate gene expression…… whose ability? Please write a concise title which is understandable for the readers.
Response 1. We have shortened the title trying to highlight all the distinctive aspects of this work.
Point 2. Introduction: Page 1, line # 40. Many industrially used chemicals, if not managed properly, escape from various plastic products and waste, spreading into the environment. Re-write this sentence clearly and avoid the excessive use of commas.
Response 2. The authors agree with the referee, the sentence was very confusing and not easy to understand. The sentence has been rewritten more concisely.
Point 3. Page 2, line # 80-81. Remove the full stop in the middle of sentence and if second sentence has started from among, then write the first word of second sentence in capital letter.
Response 3. We have totally rewritten the sentence, expanding it and adding references for better understanding.
- Page 2, line # 85-88. In this study, ---------------------
Write more detail about the current work as this is the last paragraph of introduction and it should have enough detail about the current work.
Response 4. The last paragraph of the introduction has been expanded and some details have been added to make the work more attractive.
Point 5. Experimental: Page 3, line # 104. There is a heading 2.2 Chromatographic analysis, then there is another heading 2.3.1 HPLC analysis, is there any difference? Don’t repeat the heading and include all information about HPLC analysis in the same heading chromatographic analysis
Page 3, line # 130. There is another heading of the same number 2.3.1 Preparation of standard solution, please correct the numbering of the headings.
Response 5. The authors agree with the referee. only one chapter was done with HPLC equipment and conditions to avoid repetitions and the numbers were changed in agreement with the referee.
Point 6. Results & Discussions: The resolution of Fig 1-3 is very poor and there is no text on y axis, please describe what the y axis represent?
Figure 7. HPLC-PDA chromatograms of blank extract (-----) and blank spiked extract ( ) with the 331
investigated phthalates at the concentration of 0.75 µg/mL.
Write (a) and (b) in the above caption of Fig 7.
Response 6. The authors apologize for the poor resolution, in the new version we have tried to increase it and in agreement with the referees all the X and Y axes have been made explicit. Furthermore, in accordance with the suggestion of the reviewer whom we thank, the caption of figure 7 has been modified.
